# Gut Microbiota Mediates Skin Ulceration Syndrome Outbreak by Readjusting Lipid Metabolism in *Apostichopus japonicus*

**DOI:** 10.3390/ijms232113583

**Published:** 2022-11-05

**Authors:** Zhen Zhang, Mingshan Song, Zhimeng Lv, Ming Guo, Chenghua Li

**Affiliations:** 1State Key Laboratory for Managing Biotic and Chemical Threats to the Quality and Safety of Agro-Products, Ningbo University, Ningbo 315211, China; 2Laboratory for Marine Fisheries Science and Food Production Processes, Qingdao National Laboratory for Marine Science and Technology, Qingdao 266071, China

**Keywords:** *Apostichopus japonicus*, gut microbiota, lipid metabolites, skin ulceration syndrome

## Abstract

The intestinal tract is the most important location for symbiotes and pathogens, and the microbiota plays a crucial role in affecting the health of the gut and other host organs. Dysbacteriosis in the intestinal system has been proven to be significant in skin ulceration syndrome (SUS) in sea cucumbers. This study investigates whether the gut microbiota and lipid metabolites are relevant to the initiation and progression of SUS in a *Vibrio-splendidus*-infected sea cucumber model. The tight junction genes were downregulated and the inflammatory factor gene transcriptions were upregulated after *V. splendidus* infection in the intestinal tissue of the sea cucumber. *V. splendidus* infection modulated the gut microbiota by interacting with *Psychromonas macrocephali*, *Propionigenium maris*, *Bacillus cereus*, *Lutibacter flavus,* and *Hoeflea halophila*. Meanwhile, the metabolites of the long-chain fatty acids in the intestinal tissue, including triglycerides (TG), phosphatidylethanolamines (PE), and phosphatidylglycerols (PG), were altered after *V. splendidus* infection. *V. splendidus* engaged in positive interactions with PG and PE and negative interactions with specific TG. These results related to gut microbiota and metabolites can offer practical assistance in the identification of the inflammatory mechanisms related to SUS, and this study may serve as a reference for predicting the disease.

## 1. Introduction

The gut microbiota is a crucial component of the intestinal microenvironment and interacts with the host’s physiological state, immune regulation, and dietary conditions [1,2]. An imbalance in the intestinal flora has been associated with diverse physiological functions of the host, such as colitis and other immune and metabolic diseases [3,4], and is also causally linked to host metabolism disorders. Evidently, gut microbial disorders and imbalances produce passive effects on the host’s health [5]. One of the most significant elements affecting diseases in aquatic animals is pathogen infection [6]. Alterations in the complex gut microbial community due to the invasion of pathogens usually causes significant disturbances in the structure and diversity of the gut microbial community and influences the health of the host. Colonization in the intestinal mucosa and/or lumen is the essential step in the pathogenesis of enthetic pathogenic bacteria. Pathogenic bacteria must establish themselves in a competitive environment that includes the complicated microflora that is already present [7]. An increasing number of diseases are regarded as the results of infections by a variety of pathogenic bacteria (i.e., polymicrobial diseases) [8] or an overall imbalance in the microflora and microenvironment (i.e., dysbiosis) [9]. Black band disease in corals has proven to be a model polymicrobial disease that is caused by a variety of microbial infections [10]. Meanwhile, several chronic human diseases are accompanied by microbial dysbiosis, including ulcerative colitis, allergies, rheumatic arthritis, and Alzheimer’s disease [9].

Although host and gut microbiota metabolic activities can occur simultaneously, the host relies on its microbiome to collect enzymes for digestion and metabolism [11,12]. Recent studies have shown that an unbalanced microflora, along with its related metabolites, such as short-chain fatty acids [13], and other metabolic disorders may be related to various diseases, including metabolic perturbation and inflammatory bowel disease [14,15]. Most studies on gut microbiota and metabolic disorders have focused on short-chain fatty acids. Studies on long-chain fatty acids have rarely been reported. Lipid is an essential component of cell membranes and is heavily involved in energy storage. It thus plays a vital role in many significant metabolic activities. The gut microbiota can influence the central and peripheral lipid metabolism of the host significantly [16]. Studies have suggested that lipid metabolism might be a crucial method by which gut microbiota affect the host’s health and physiological activity. Prior research has shown that an unbalanced gut microbiota has a positive effect on lipid peroxidation and triglyceride storage [17], and the significantly differentially expressed genes in germ-free mice were mainly associated with steroid and lipid metabolism [18,19]. The previous studies showed that *Blautia*, *Faecalibacterium*, and *Ruminococcus* were the major maleficent bacteria in Crohn’s disease and ulcerative colitis [20,21], and it was discovered that *Clostridium immunis*—when administered to colitis-prone mice—protected them against colitis-associated death [22]. In addition, studies have pointed to the roles of several bacteria in colorectal carcinogenesis, including *Fusobacterium nucleatum* and certain strains of *Escherichia coli* and *Bacteroides fragilis*, and reported associations between bacterial markers and treatment efficacies or clinical outcomes, increasing the potential of these markers to be used for treatment prediction and prognostication [23,24]. The need to reveal the characteristics of the interactive patterns of the microbiota that lead to disease occurrence and development is becoming increasingly urgent as epidemics rise in frequency [25], particularly among marine animals. *Apostichopus japonicus* (Echinodermata, Holothuroidea) is a fishery species that is of economic importance in Asian countries [26]. However, they regularly experience outbreaks of skin ulceration syndrome (SUS), which is caused by bacterial infections, particularly infections involving *Vibrio splendidus* [27]. Nevertheless, no studies have been published that focus on detecting the changes in the gut microbiota and metabolic patterns before SUS, which may be regarded as biomarkers that can be used to predict or prevent disease development.

Microbes play a critical role in host health and disease appearance. In view of this, the main purpose of this study was to expose the potential characteristics of sea cucumber microbiota during the pathogenesis and occurrence of SUS. We performed time course experiments on the *A. japonicus* breed in an aquaculture pond as they naturally developed SUS and compared the gut microbial community with that of *A. japonicus* whose health was maintained under the same conditions. This experimental design allowed us to measure the inter-individual microbiota variation and to examine a number of hypotheses, as follows: (1) microbial community differences exist between SUS and healthy *A. japonicus*, (2) the bacterial community’s remodeling and eventual simplification occur as the disease progresses, and (3) a time course experiment on healthy and SUS animals can reveal the extent of the disease caused by putative pathogens and metabolites. Together, these findings reveal distinct categories of dysbiosis, which can inform the development of microbiome-based diagnostics and therapeutics.

## 2. Results

### 2.1. The Intestine Histology and Characteristics of A. japonicus under V. splendidus Infection

The histopathological experiment showed that the villi cells in the control group were elongated and finger-shaped, with a good morphology and an unbroken structure. The epithelial cells were neat and clean, the striatum was intact, and the goblet cells were sparsely distributed (Figure 1A). In the SUS group, the intestinal tissue was aneretic, with a sparse striated border, ruptured villi, disordered and loose epithelial cells, and mild migration of some epithelial cells (Figure 1B). 

Given that pathogens readily invade the submucosa and vasculature when the permeability of the intestinal mucosa increases, thereby accelerating the inflammatory response, we next sought to identify the junctions of the intestinal tissue. Changes in the expression levels of ZO-1 and occludin, representative markers for tight junctions, were assessed by qPCR. The expressions of AjZO-1 and Ajoccludin at the mRNA level were significantly decreased after 12 h in *V. splendidus*-challenged sea cucumber (Figure 1). AjZO-1 and Ajoccludin were downregulated by 0.09-fold (Figure 1C) and 0.03-fold (Figure 1D) in the SUS group compared with the normal group, respectively.

We investigated the intestinal tissue inflammatory factors of interleukin (IL)-17, p105, transforming growth factor-β (TGF-β), the vascular endothelial growth factor (VEGF), and macrophage migration inhibitory factor (MMIF). The expressions of IL-17, p105, TGFβ, VEGF, and MMIF were increased significantly after *V. splendidus* infection (*p* < 0.05) (Figure 1E). The expressions of IL-17, p105, TGFβ, VEGF, and MMIF were increased by 145.11-fold, 6.32-fold, 83.46-fold, 2.86-fold, and 10.12-fold, respectively, in the SUS group when compared with the control group (*p* < 0.05). 

### 2.2. Overview of the Microbiota Composition of the Intestinal Contents of A. japonicus with V. splendidus Infection

This section concerns the identification of the intestinal microbial community in sea cucumber. In order to investigate the changes in the microbial communities in sea cucumber at different times after *V. splendidus* infection, we sequenced 16S rRNA genes of the intestinal microbes in the sea cucumber. We constructed 16S rRNA libraries, and they were sequenced using an Illumina MiSeq sequencer. The α-diversity index was applied to assess the intestinal microbial diversity of the sea cucumber. The Shannon diversity index was significantly decreased 48 h after *V. splendidus* infection (Figure 2A). The Observed_species and Chao1 diversity indexes showed no significant differences between the groups (Figure 2B,C). Moreover, based on the OTUs detected in the five groups, we found that the NMDS ordination displayed significant differences in the intestinal microbiota between the different stages of *V. splendidus* infection, and the intestinal microbiota showed a certain substitution and evolution trend (Figure 2D). This model was further verified via an analysis of similarity (ANOSIM) test based on the Bray–Curtis distance, which showed that, except for the SUS group and 96 h interval (*p* = 0.0521), there were significant differences in the bacterial communities between the different groups (Appendix A). Moreover, the magnitude of the differences in the intestinal microbiota consistently increased (r = 0.8415, *p* < 0.001) 48 h after *V. splendidus* infection (Appendix A). In other words, the intestinal microbiota tended to differ more significantly with time following *V. splendidus* infection. A total of 6399 total bacterial OTUs were detected, with a 97% sequence similarity. Among them, 723 OTUs were identified in all five segments, and 1502, 904, 732, 342, and 272 OTUs were obtained after 0 h, 12 h, 48 h, and 96 h and from the SUS group, respectively (Figure 2E). This shows that the bacteria numbers were lower after *V. splendidus* infection in this group than in the control group.

### 2.3. V. splendidus Infection Altered the Major Microbiota Abundance of A. japonicus

The core intestinal microbiota plays a crucial role in several physiological changes in the host. The eight most abundant phyla (in decreasing order) were Proteobacteria, Firmicutes, Actinobacteria, Bacteroidetes, Verrucomicrobia, Chloroflexi, Acidobacteria, and Fusobacteria, which accounted for 61.55%, 14.57%, 9.57%, 8.46%, 3.08%, 0.82%, 0.34% and 0.10% of the total microbial community, respectively. Together, they represented 98.50% of the total bacteria in the control group. The same major phyla were found in the other groups, but the abundance of the microbiota was different (Appendix A). The abundance of Proteobacteria significantly decreased from 61.55% ± 8.53% in the control group to 45.90% ± 5.44% in the 48 h group and recovered to 74.80% ± 14.79% in the SUS group. The relative abundance of Firmicutes showed the opposite trend; it was 14.57% ± 10.79% in the control group and 38.21% ± 6.42% in the 48 h group, but then it decreased to 8.74% ± 1.31% in the 96 h group. Meanwhile, the Actinobacteria abundance decreased to 2.26% ± 1.65% in the SUS group from 9.57% ± 1.50% in the control group (Figure 3A, Appendix A). Interestingly, the ratio of Firmicutes to Bacteroidetes (F/B) was significantly higher (5.44 and 4.06) in the 12 h and 48 h groups than in the control group (1.72) (Appendix A). This is in good agreement with several studies on rodents and humans, showing that a high F/B ratio is associated with obesity and inflammation [28]. At the level of fine bacterial families, *V. splendidus* infection resulted in a significant increase in the Rhodobacteraceae abundance (47.81% ± 21.88%) in the SUS group compared with the control group (19.76% ± 6.52%), and the Desulfobulbaceae abundance was decreased from 5.17% ± 3.65% in the control group to 0.53% ± 0.52% in the SUS group (Appendix A). It is worth noting that the Bacillaceae abundance increased 48 h after *V. splendidus* infection to 34.40% ± 6.78% and returned to 4.56% ± 1.09% in the 96 h group (Figure 3B, Appendix A). 

Meanwhile, Actinobacteria species, such as *Ilumatobacter fluminis* and *Rhodococcus qingshengii*, and Proteobacteria species, such as *Sulfurovum lithotrophicum*, *Glaciecola amylolytica*, *Kineobactrum sediminis*, and *Acinetobacter johnsonii*, decreased as the *V. splendidus* infection progressed. Furthermore, the *V. splendidus* infection modulated the gut microbiota by increasing the abundances of *Pelagimonas varians*, *Psychromonas macrocephali*, *Hoeflea halophila*, and *V. splendidus* and influencing the abundance of *Bacillus cereus*. The abundance of *V. splendidus* significantly increased in the 12 h group (2-fold) and increased by 5-fold in the 48 h and 96 h groups compared with the control group (Figure 3C).

### 2.4. Exploring the Interspecies Interactions of the V. splendidus Pathogen

To reveal which resident taxa activate or repress the pathogen, we focused on the modules of *V. splendidus*. We identified 13 OTUs that interacted directly with *V. splendidus* during the infection (Figure 4). The types of interaction between the core OTUs and *V. splendidus* exhibited a distinct distribution. For example, *Psychromonas macrocephali, Propionigenium maris, Bacillus cereus, Lutibacter flavus,* and *Hoeflea halophila* reacted positively to *V. splendidus* infection, whereas *Aliiruegeria haliotis*, *Sulfurovum lithotrophicum*, *Acinetobacter johnsonii, Glaciecola amylolytica, Rhodococcus qingshengii*, and *Desulfocastanea catecholica* reacted negatively to *V. splendidus* during the process of the *V. splendidus* infection of the sea cucumber (Figure 4). *Psychromonas macrocephali, Hoeflea halophila,* and *Bacillus cereus* had higher abundances than the bacteria with negative networks.

### 2.5. V. splendidus Infection Changed the Lipid Metabolism in the Sea Cucumber Gut Contents

The samples were analyzed under LC–MS positive and negative ion detection modes. Since not all lipids can be efficiently ionized in one mode, positive and negative ion electrical pulses were used. The mass spectral data were processed by multivariate analysis. Approximately 947 peaks of positive ions (ESI^+^) and 441 peaks of negative ions (ESI^−^) were detected by Micromass Marker Lynx when using the same acquisition method from 0 to 34 min of retention time. We normalized the peak value and analyzed the pareto-scaled data through PCA, PLS-DA, or OPLS-DA. PCA is an unsupervised clustering method, which is used to reveal the system trend and ‘clustering’ in multivariate data. The PCA score plots highlight the formation of clusters or patterns two-dimensionally, thus showing the similarities and differences between the samples. PCA was used to analyze the intestinal lipid metabolism profile to reveal the effects of *V. splendidus* infection on the intestinal lipid metabolism of the sea cucumber. In the ESI^+^ and ESI^−^ modes, the cluster formation of the first two components was time-dependent, which explained 62.5% and 51.9% of the total variance, respectively (Figure 5). The further model validation using PLS-DA shows that the permutations are equal to 200, generating intercepts with R2 = 0.372 and Q2 = −0.178. Together with the results obtained in the negative mode, these results show that the OPLS-DA model was well-suited and statistically valid. The same conclusion was reached when using the supervised projection method, PLS-DA. The first two components were observed at different times of the *V. splendidus* infection and in the control groups in both the ESI^+^ and ESI^−^ modes. 

The S-plot shows the covariance and correlation between the metabolites and can thus be used to identify the pivotal metabolites [29]. The significant metabolites at the top and bottom of the S-plots (Appendix A) were associated with group separation. Based on the parameters of variable importance in prediction (VIP), the top 25 lipid metabolites in both the ESI^+^ and ESI^−^ patterns served as potential biomarkers (Appendix A). The levels of triglycerides (TG) (18:1/18:2/18:2; 18:2/18:2/18:2; 18:0/16:0/20:4; 18:1/18:1/18:2) increased 12 h after *V. splendidus* infection, and the levels of TG (16:0/16:0/18:1; 16:0/14:0/18:1) decreased 48 h after *V. splendidus* infection. Furthermore, the phosphatidylethanolamine (PE) (16:0/16:1) and phosphatidylglycerol (PG) (16:0/16:1; 16:0/18:2) levels were increased in the 96 h and SUS groups after *V. splendidus* infection, and the phosphatidylcholine (PC) levels (18:1/20:4; 18:1/20:5) were decreased after *V. splendidus* infection (Figure 6).

### 2.6. The Relationship between the Gut Microbiota and Lipid Metabolite Profiles

Mantel tests were conducted to detect the correlations between the gut microbiota and lipid metabolite profiles. The correlations between some specific gut microbes and lipid metabolites are shown using an interaction network in Figure 7. For example, *Desulfocastanea catecholica*, *Rhodococcus qingshengii*, *Aliiruegeria haliotis*, *Glaciecola amylolytica*, and *Sulfurovum lithotrophicum* were negatively correlated with long-chain fatty acids, such as TG (18:1/18:2/18:2; 18:2/18:2/18:2; 18:1/18:1/18:2; 18:0/16:0/20:4), PG (16:0/16:1; 16:1/18:2) and PE (16:0/16:1), but showed positive correlations with TG (16:0/16:0/18:1; 16:0/14:0/18:1; 16:1/16:1/20:5), SQMG (16:0), and PC (18:1/20:4). Interestingly, the OTUs assigned to *Psychromonas macrocephali, Bacillus cereus, Lutibacter flavus, Hoeflea halophila*, and *Pelagimonas varians* showed contrasting correlations with the long-chain fatty acids. *V. splendidus* exhibited a positive interaction with PG (16:0/16:1; 16:1/18:2) and PE (16:0/16:1; 16:0/18:1) and a negative interaction with specific TG (Figure 7).

## 3. Discussion

The intestinal microbiota has been reported to play an active role in maintaining intestinal homeostasis [30]. This homeostasis is closely related to a series of biological processes of the host, such as innate immunity, defense against pathogens, digestion, and intestinal function maturation, in a wide variety of animals [3,31]. Nevertheless, the intestinal microbiota’s role in mediating metabolites and the host immune system upon infection with pathogenic bacteria in sea cucumbers remains to be determined. The intestinal barrier protects the host from intestinal microbes and gastrointestinal toxins. The destruction of the intestinal barrier can result in the entry of microbial components into the host, generating systemic low-grade inflammation [32]. The intestinal barrier cells are held together with junctions (ZO-1, claudin, and occludin) to prevent the easy invasion of potential pathogenic bacteria. The epithelial surface is covered with a mucus layer, which protects the intestinal tract from harmful microbes and substances. In a shrimp intestinal inflammation model, the intestinal tract damage was serious, and almost all the intestinal villi fell off or were non-existent. The mucosal structure was loose, and the submucosa partially dissolved. *Vibrio parahaemolyticus* can induce a significant intestinal inflammatory response in shrimp and can significantly upregulate the expression of the inflammatory cytokine TNF-α, Ras-related protein Rab6A (RAB6A), and lipopolysaccharide-induced TNF-α factor (LITAF) [33]. To better understand the relationship between intestinal inflammation and pathogen infection, we investigated the intestinal inflammation characteristics of *A. japonicus* during the occurrence of SUS induced by *V. splendidus*. We found that the intestinal tissue was damaged, the stripe boundary was sparse, and the villi were broken, and we also noticed changes in the number of inflammatory cells that had infiltrated the intestinal tissue (Figure 1A,B). AjZO-1 and Ajoccludin were downregulated (Figure 1C,D) in the SUS group when compared with the control group. Moreover, the intestinal tissue inflammatory factor levels of IL-17, p105, TGF-β, VEGF, and MMIF were increased significantly after *V. splendidus* infection (*p* < 0.05) (Figure 1E). This indicates that the SUS disease caused by *V. splendidus* involved the development of intestinal inflammation.

Accumulating evidence has demonstrated that the intestinal microbiota plays an essential role in the development and evolution of intestinal inflammation in vertebrates. A series of data from animal models and clinical studies have confirmed the dysbiosis of the gut microbiota during intestinal inflammation [34], which manifests itself in the decreased diversity, irregular composition, and changes in the spatial distribution and/or function of the gut microbiota [35]. Microbial diversity was remarkably downregulated in a diseased grass carp (*Ctenopharyngodon idellus*) [36] and an adult turbot (*Scophthalmus maximus*), and this process is usually gradual [37]. In the SUS model of *V. splendidus* infection in the sea cucumber, the diversity was significantly downregulated according to the Shannon diversity index analyzed, but the Observed_species and Chao1 diversity indexes were not significantly different (Figure 2). That is, the intestinal microbiota generally tended to differ more significantly with time after *V. splendidus* infection, and *V. splendidus* is the dominant bacterium that causes SUS in sea cucumber. In particular, the reduced diversity of the fecal microbiota is the most consistent feature of intestinal inflammation [38]. For example, ulcerative colitis (UC) patients show fewer Firmicutes and a higher abundance of Bacteroidetes [28] at the phylum level compared to healthy individuals. In a model of intestinal inflammation in sea cucumber caused by SUS, the bacterial abundance of the two phyla also showed a similar trend to the UC model (Figure 3). During intestinal inflammation, not only is the diversity of the microbiota and communities reduced, but the levels of maleficent bacteria are also increased. Intestinal inflammatory diseases are usually associated with changes in the intestinal microbiota, including the expansion of the proteus facultative anaerobic bacteria of the Proteobacteria phylum, which is a common feature of biological disorders [39]. Therefore, the increasing phyla are mainly Proteobacteria, which shows that 14 of the 24 differential bacteria examined in our study belong to the Proteobacteria phylum (Figure 3). Although the abundance of the bacteria during intestinal inflammation was similar to the UC pattern, there was an obvious increase in the numbers of Firmicutes and Bacteroidetes during the *V. splendidus* infection. These may be involved in the process of resisting excessive inflammation. Above all, intestinal microbial linkage caused by *V. splendidus* infection alters the structural layout of gut microbes, leading to eventual intestinal inflammation and SUS. 

*V. splendidus* is a Gram-negative bacterium that is widespread in marine coastal environments [40]. *V. splendidus* is associated with the mortality of scallops (*Argopecten purpuratus*) [41], oysters (*Crassostrea gigas*) [42], turbots (*Scophthalmus maximus*) [43], and other marine animals worldwide. During infection, the pathogens are closely related to the microbial species that colonize the intestine. The interaction of the pathogens with both the host cells and the intestinal microbiota is a critical factor that affects the host’s health. Pathogens interact with symbiotic bacteria or potential pathogens to produce metabolites (such as bile acids and short-chain fatty acids) by regulating the intestinal mucosal immunity and have an impact on the intestinal ecosystem. *V. splendidus* is positively correlated with *Psychromonas Antarctica, Propionigenium maris*, and *Bacillus cereus* during infection (Figure 4). In sea urchins, propionic acid was found to be a microbial metabolite that promoted health. It can promote the production of organisms when they are in adverse environments [44]. *P. maris*, one of a large number of *Propionigenia*, may be an important microbial defender against crises through propionate production. *B. cereus* is associated with food poisoning, which is often related to a diet that contains rice-based dishes. The infected organism develops emetic and/or diarrheal syndrome caused by an emetic toxin and enterotoxin. It also produces other toxins, such as phospholipases, proteases, and hemolysins. These toxins may be essential for the pathogenicity of *B. cereus* in non-gastrointestinal diseases [45]. In our study, not only did we find bacteria that were positively correlated with *V. splendidus*, but we also found some that were negatively correlated with it, such as *Sulfurovum lithotrophicum*, *Desulfocastanea catecholica*, and *Acnetobacter johnsonii*. *Sulfurovum lithotrophicum* and *Desulfocastanea catecholica* were associated with mesophilic sulfur and thiosulfate oxidization for the production of energy via the metabolism [46,47], and *A. johnsonii* is widely distributed in natural environments, even though it rarely causes infections [48]. Most of the bacteria that were negatively correlated with *V. splendidus* were related to the energy metabolism competition and utilization of *V. splendidus.* To sum up, *V. splendidus* may ultimately affect intestinal health and cause SUS through interactions with different bacteria.

As we know, lipids are the important and most abundant components of living systems. Lipid metabolism disorders are related to many diseases, such as atherosclerosis, metabolic disorders, cardiovascular diseases, cancer, infections, and inflammatory diseases. Recent studies have linked lipidomic alterations to different cellular fates and have shown that lipids display distinct functions according to their chemical specificity [49]. The complex gut microbiota profoundly affects the host’s gut health and susceptibility to systemic disease. As part of their functional mechanisms, these symbiotic bacteria secrete bioactive lipid mediators, leading to the accumulation of intestinal major lipids, phospholipids (PC and PE), and TG, which promote healthy gut development and contribute to the prevention of intestinal infections [50,51]. Using untargeted lipidomics, we found that the major fatty acids in the TGs of all the specimens were 16:0, 18:0, 18:1, 18:2 (LA, linoleic acid), 20:5 (EPA, icosapentaenoic acid), and 20:4 (AA, arachidonic acid). The main different chain fatty acids were 18:1, 18:2, 20:4, and 20:5 in the case of the PC contents, and the main different chain fatty acids were 18:1 and 18:2 of PE accumulated during *V. splendidus* infection. Some of these long-chain fatty acid chains (e.g., 20:4 and 20:5) are synthesized from precursor essential fatty acids (LA and α-LA, respectively) and are then assembled into TG, PC, or PE. These abovementioned lipids can directly participate in inflammation responses [52,53]. In humans, the levels of PE and PG in the chronic urticaria (CU) were significantly increased, while the levels of phosphatidylcholine (PC) were significantly decreased [54]. PC, known as anti-inflammatory or antioxidant phospholipids, are involved in membrane structure formation and cell signaling [55]. PC can ameliorate experimental arthritis and LPS-induced inflammation by regulating leukocyte activation and the intestinal–brain axis balance [54]. During our study, the AA-PC and EPA-PC were downregulated after *V. splendidus* infection at 48 h and 96 h, and they subtly increased in the SUS model. At the same time, the PG and PE were increased after the *V. splendidus* challenge. Meanwhile, the TGs with branched-chain amino acids (18:2, 20:4) that can regulate inflammation were also significantly increased (Figure 6). In the process of inflammation induced by pathogen infection, there is a process from acute inflammation to chronic inflammation similar to that of higher animals. Sudden 12 h stimulation leads to the upregulation of inflammation-related substances in order to clear the bacteria. After 48 h, organism resistance leads to the bacteria to become non-dominant. At 96 h, the organism cannot maintain bacterial resistance and develops chronic inflammation or traumatic inflammation. For example, TGs (18:1/18:2/18:2; 18:2/18:2/18:2; 18:0/16:0/20:4; 18:1/18:1/18:2) promote inflammation in the branched-chains, which induce inflammation when upregulated, but when downregulated, some branched-chain lipids provide basic energy or inhibit inflammation, such as the TGs (16:0/16:0/18:1; 16:0/14:0/18:1), showing the same change trend as that of the intestinal microbiota in the process of inflammation. Upon pathogenic stimulation, hydrolyzation produces free fatty acids to regulate inflammation. Many previous studies have shown that disorder in the gut microbiota can disturb lipid peroxidation and the regulation of metabolites [17], affecting the host’s health. *Bifidobacterium pseudolongum* significantly decreased the triglyceride levels and showed potential benefits in obese individuals [56]. In addition, we constructed a network of interactions between the intestinal microorganisms and lipid metabolites during *V. splendidus* invasion, and it is clear that upregulated or inhibited lipid metabolites play an important role in the pathogenesis. 

## 4. Materials and Methods

### 4.1. Ethics Statement

The sea cucumbers used in this study were animals cultured in a commercial market, and all the experiments involving the animals were managed with reference to the criteria set out in the Guide for the Care and Use of Laboratory Animals of the National Institutes of Health. The experimental protocol was authorized by the Experimental Animal Ethics Committee of Ningbo University, China.

### 4.2. Animal Collection and Experimental Design

Healthy sea cucumbers averaging 120 ± 10 g in weight were acquired from CenShi Sea Products Co., Ltd. (Dalian, China), where no epidemic diseases were reported to have occurred. The animals were fed daily and given one week to acclimatize to the holding ponds with 1000 L of seawater before bacterial infection. Each pond was aerated 24 h a day via five aeration stones.

*V. splendidus* was cultured on 2216E media and proliferated to OD_600_ = 1.0. The bacterial suspensions were centrifuged at 4500× *g* for 15 min, the 2216E media was removed, and then they were suspended in seawater and diluted to 1 × 10^7^ CFU/mL. The infection experiment was carried out using eighty sea cucumbers in an aquaculture pond with 1000 L of seawater in each group.

The sea cucumbers were obtained at 0 h, 12 h, 48 h, and 96 h post-inoculation with *V. splendidus*, and the SUS individuals had white anabrosis on their body walls. The SUS sea cucumbers were taken out of the experimental pond as soon as they were observed. We collected eight samples from each experimental group. The intestinal tissue of the individual was taken out, and the intestinal contents were transferred via lancet for storage in 2 mL sterile plastic cryotubes under sterile conditions before immediately being stored in liquid nitrogen.

### 4.3. The Effects of V. splendidus Infection on the Sea Cucumber Intestine Histology

After preparation, the intestinal tissues were fixed in 2.5% glutaraldehyde (4 °C) for 24 h, washed several times in PBS (0.1 M, pH 7.2–7.4), and dehydrated with incremental levels of alcohol (30%, 50%, 70%, 80%, 90%, and 100% × 2; 10 min for each step). The dehydrated specimens were immersed in paraffin and cut into 5 μm-thick slices. The slices were dyed using hematoxylin/eosin (H&E) via our standard laboratory protocol. The H&E figures were obtained using a Zeiss Axio Scope A1 microscope and an AxioCam MRc5 digital camera. 

### 4.4. The Effects of V. splendidus on the Immune Genes

Intestinal tissue from eight randomly selected sea cucumbers from both the control and the *V. splendidus*-affected groups was prepared for total RNA extraction to detect the effects of the *V. splendidus* infection on the immune-related genes (*n* = 8). The primers were designed using Primer 5 software (Applied Biosystems, Waltham Mass, USA). Sets in the efficiency range of 90–110% were performed for qRT-PCR. Total RNA was extracted using the TransZol Up Plus RNA Kits, and then the cDNA was synthesized using TransScript All-in-One First-Strand cDNA Synthesis SuperMix Kits (AT341, TransGen Biotech, Beijing, China). The qRT-PCR was carried out using a Rotor-Gene 6000 real-time PCR instrument (Corbett Robotics Inc., Sydney, Australia) with SYBR^®^ Premix Ex TaqTM II based on the manufacturer’s protocol. Each reaction was measured five time at a volume of 20 microliters (10 μL of SYBR mixture, 1.6 μL of 10 μM primers, 2 μL of cDNA, and 6.4 μL of RNase-free ddH_2_O). The qRT-PCR method was performed as follows: 95 °C for 10 min, 40 cycles of 95 °C for 10 s, 55 °C for 10 s, and an extension at 72 °C for 20 s. The quantified qRT-PCR products were calculated via the 2^−ΔΔCT^ method [57]. The primer sequences referred to in this study are shown in Appendix A.

### 4.5. The Gut Microbiome Analysis

The intestinal contents’ genomic DNA (gDNA) was obtained using a Fast DNA^®^ Spin kit (Bio 101, Carlsbad, CA, USA) according to the manufacturer’s protocol. The quality and concentration of the gDNA were detected using a Nano Drop ND-2000 spectrophotometer (Nano Drop Technologies, Wilmington, NC, USA) and 1.2% agarose gel electrophoresis. To amplify the bacteria, the V3–V4 regions of the 16S rRNA gene, with the specialized primers 338F (5′-GTACTCCTACGGGAGGCAGCAG-3′) and 806R (5′-GGACTACHVGGGTWTCTAAT-3′), were used according to the method described by Kozich et al. [58]. PCR was carried out under the following conditions: the first step was denaturation at 95 °C for 30 s, followed by annealing at 55 °C for 30 s, an extension step at 72 °C for 45 s × 25 cycles, and a final extension at 72 °C for 10 min. The PCR products were measured by electrophoresis on 1.2% agarose gel to ensure that they only contained one band and were equal in size. The amplicons were purified using a PCR fragment purification kit, and their concentration was measured using a PicoGreen-iT dsDNA Assay Kit (Invitrogen, Carlsbad, CA, USA). For the PCR products, the DNA in the amplicons was sequenced using an Illumina MiSeq platform (Illumina, San Diego, CA, USA), which conducted 2 × 300 bp paired-end reads.

The paired-end reads were assigned to the specimens according to their unique barcodes, and we cut off the primers and barcodes and then combined them using FLASH [59]. The 16S rRNA sequences eliminated the equivocal bases and truncated any site of more than three subsequent bases, and then deleted the chimera sequences and singletons by comparison with the reference database (Gold Database) via the UCHIME algorithm [60]. The high-quality 16S rRNA sequences were obtained using the Quantitative Insights Into Microbial Ecology pipeline (QIIMEv1.9.0) [61]. The sequences with a ≥ 97% similarity were assigned to the same operational taxonomic units (OTUs), and bacteria phylotypes were labelled using the UCLUST method [62]. The most abundant sequence in a given OTU was taxonomically annotated as the representational sequence for the next analysis. Then, the OTUs that belonged to the databases of the archaea, chloroplasts, and eukaryota were removed. The next representative sequences were annotated. The representative sequence was the closed Green Genes Database, annotated using PyNAST. The alpha diversity was used to analyze the species diversity complexity, and the alpha diversity index was computed using QIIME. Changes in the diversity of the bacterial community were compared between the specimens using one-way analysis of variance (ANOVA) and Tukey’s post hoc test. We assessed all the differences in the bacterial community structure via non-metric multidimensional scaling (NMDS) and analysis of similarity (ANOSIM) [63]. The microbial community at the phylum and family levels were shown by stacked columnar charts using OriginPro^@^2018. Available online: http://www.originlab.com (accessed on 6 January 2018). The indicator values (IndVal) were used to recognize the indicator bacteria associated with the progress of the disease. The indicators were shown on a heatmap created by the package “labdsv” in R V3.0.2 [64].

### 4.6. High-Performance Liquid Chromatography–Tandem Mass Spectrometry (HPLC–MS/MS) Assay

The total lipid was extracted using the chloroform–methanol method. In brief, 50 mg of intestinal contents (freeze-dried) was obtained using a chloroform/methanol/water (1:2:0.8, *v*/*v*/*v*) solution that included 0.05% 2,6-di-tert-butyl-4-methylphenol (BHT), after which the total lipid was isolated by ultrasonication for 15 min and centrifugation for 10 min. The lipid extract was evaporated using a rotary evaporator, and the residue was stored at −20 °C for the next analysis. The total lipids were dissolved in 1000 μL of methanol and then centrifuged at 4 °C (12,000 rpm, 10 min). The dissolved lipid was filtered using a 0.22 μm ultrafiltration membrane (Millipore, Massachusetts, USA).

The aliquot (3 μL) was injected into a reverse-phase Hypersile Gold C18 (100 mm × 2.1 mm, 1.9 μm, Thermo Fisher, USA) analytical column in a Thermo Fisher U3000 High Performance LC system (HPLC) at the Ningbo Institute of Oceanography. The elution used a mobile phase A of acetonitrile/water (6:4, *v*/*v*) and mobile phase B of isopropanol/acetonitrile (9:1, *v*/*v*) at a flow rate of 0.2 mL/min. The electrolytes were 0.1% formic acid and 10 mM ammonium acetate in the mobile phase.

The program of the gradient elution was 0–15min, 60–45% A; 15.0–18.0 min, 45–35% A; 18.0–26.0 min, 35% A; 26.0–28.0 min, 35–0% A; 28.0–30.0 min, 0% A; 30.0–30.5 min, 0–60% A; 30.5–40 min, 60% A. The temperature of the sample chamber was set at 4 °C, and the column temperature was 45 °C. Mass spectrometry was carried out using a Thermo ScientificTM Q Exactive hybrid quadrupole-Orbitrap mass spectrometer with a HESI-II probe. The instrument performed a data-dependent LC-MS/MS method in the ESI^+^ and ESI^−^ modes. The data were obtained in a centroid mode from 200 to 2000 m/z at a 70 K resolution and in an HCD MS/MS mode at a 17.5 K resolution. Then, the AGC target was set as 1e6 for MS and 2e5 for MS2. The capillary voltage was set as 3.5 kV and the capillary temperature was 320 °C. The sheath gas was 45 arb and the aux gas was 10 arb. The MS2 analysis was conducted using the mass spectrometer with a variety of collision energies and ramps of 25 and 30 V in the ESI^+^ mode and 20, 24, and 28 V in the ESI^−^ mode based on the different kinds of lipids. The spectrometer was calibrated before ESI^+^ and ESI^−^.

The original data obtained using the Thermo Xcalibur™ data processing system were managed using a Lipid searchTM4.0 system and then imported to the iOmics cloud platform or SIMCA-P software for the principal component analysis (PCA) and partial least squares discriminant analysis (PLS-DA). The variables corresponding to the specific components in the samples were selected from the loading plot. A Q-Exactive high-resolution mass spectrometer was used to examine the accurate molecular weight of compounds, and the error was within 5 ppm. LIPID databases, such as HMDB 5.0, Available online: http://www.hmdb.ca (accessed on 7 January 2022) and LIPID MAPS, Available online: http://www.lipidmaps.org (accessed on 7 January 2022), were further used to analyze the constructive structure of the lipid species.

### 4.7. Statistical Analysis

All results are presented as the mean ± standard deviation (SD). We verified whether the data had a normal distribution by Tukey’s post hoc test and a one-way analysis of variance test by SPSS, Version 19.0 (IBM, New York, USA), and *p* < 0.05 was considered significant.

## 5. Conclusions

We studied the influence of SUS progression on the intestinal microbiota and lipid metabolites of a *V. splendidus*-infected sea cucumber model. The results show that *V. splendidus* infection can regulate the intestinal microbiota by increasing the abundance of *Pelagimonas varians*, *Psychromonas macrocephali*, *Hoeflea halophila*, *Bacillus cereus*, and *Psychromonas antarctica* and by forming positive networks with these bacteria. The interaction of the major gut microbiota with lipid metabolites, such as TG and PE, boosts the SUS progression in sea cucumber. To fully understand the connections between the gut microbiota and lipid metabolites, future studies should isolate and culture a single bacterium to address this issue and determine what kinds of long-chain fatty acids are the main factors regulating inflammation. The current work can improve our current understanding of sea cucumber lipid metabolites during pathogen infections.

## Figures and Tables

**Figure 1 ijms-23-13583-f001:**
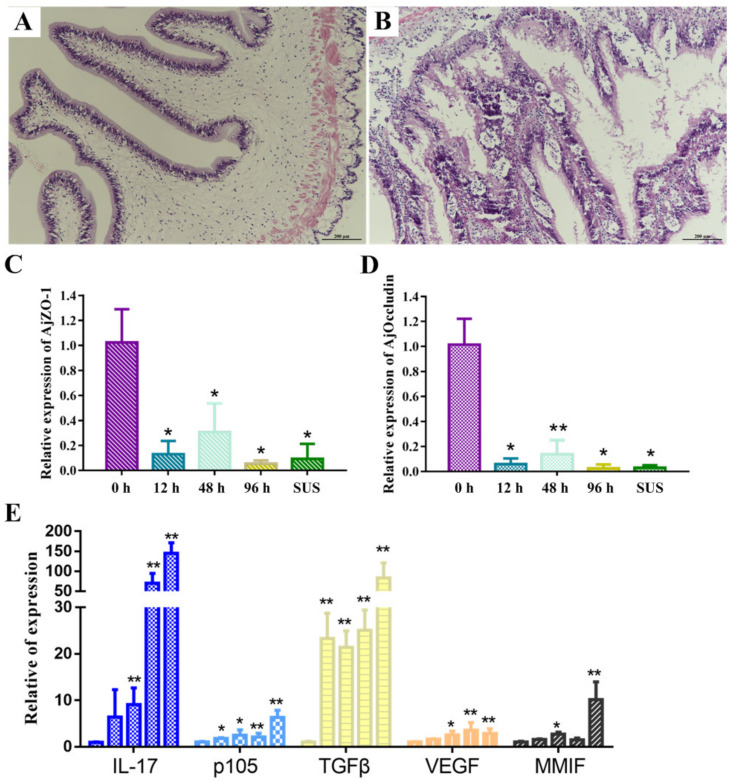
H&E staining of the intestinal tissue sections of (**A**) the 0 h group and (**B**) the SUS group. Temporal expression of tight junction proteins in *V. splendidus*-infected sea cucumber intestinal tract: (**C**) AjZO-1 and (**D**) Ajoccludin. (**E**) The expression of inflammatory factors IL-17, p105, TGFβ, VEGF, MMIF at different infectious times of *A. japonicus* infected with *V. splendidus*, with a final concentration of 1 × 10^7^ CFU mL^−1^. Asterisks indicate significant differences: * *p* < 0.05 and ** *p* < 0.01.

**Figure 2 ijms-23-13583-f002:**
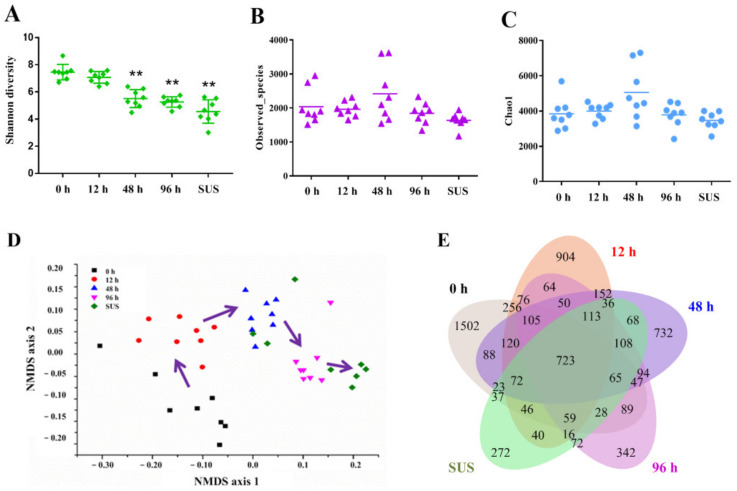
*V. splendidus* infection adjusted the gut microbiota in sea cucumber. (**A**) Shannon diversity index after 48 h groups. (**B**) Observed_species index and (**C**) Chao1 index have no significant differences. (**D**) Non-metric multidimensional scaling (NMDS) of the gut microbiota after *V. splendidus* infection. (**E**) Overlap between detected species among the different groups after *V. splendidus* infection of the sea cucumber. Asterisks indicate significant differences: ** *p* < 0.01.

**Figure 3 ijms-23-13583-f003:**
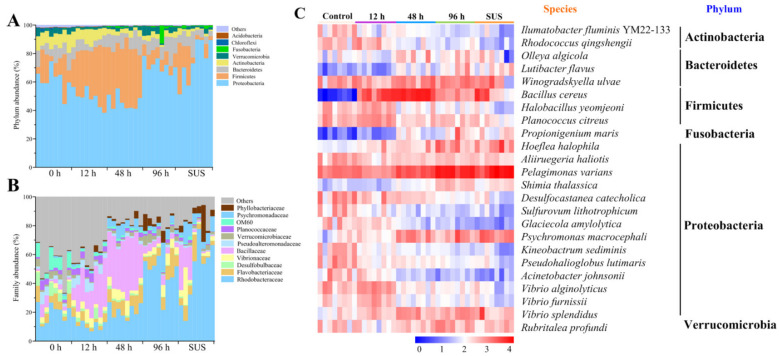
Bacterial taxonomic profiling at the (**A**) phylum and (**B**) family levels. (**C**) Heat map exhibiting the relative abundance (log2 transformed) of the 24 key species identified using the random forest method among the different groups after *V. splendidus* infection. The relative values for the bacterial species are indicated by the color intensity.

**Figure 4 ijms-23-13583-f004:**
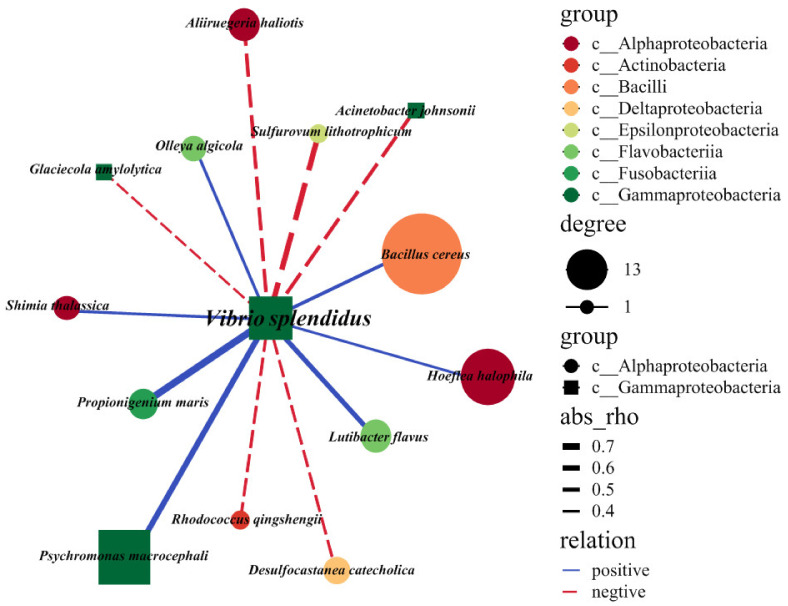
Microbial interaction networks of *V. splendidus* with other major bacterial species throughout the infection of sea cucumber. The solid (or dashed) arrows represent positive (or negative) interactions between the two individual bacteria. The sizes of the circles were proportional to the abundance of that bacteria. OTUs are colored to indicate the bacterial class level. The edge thickness is proportional to the value of Spearman’s correlation.

**Figure 5 ijms-23-13583-f005:**
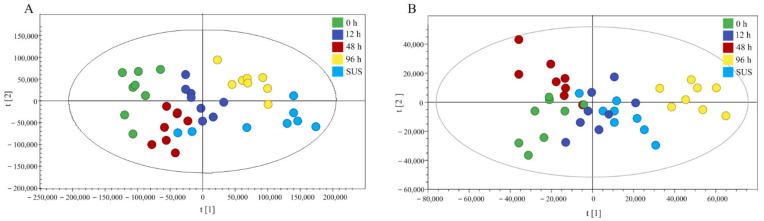
Principal component analysis score scatter plot in ESI^+^ (**A**) and ESI^−^ (**B**) modes for the total lipids in the gut of different groups of *A. japonicus* after *V. splendidus* infection.

**Figure 6 ijms-23-13583-f006:**
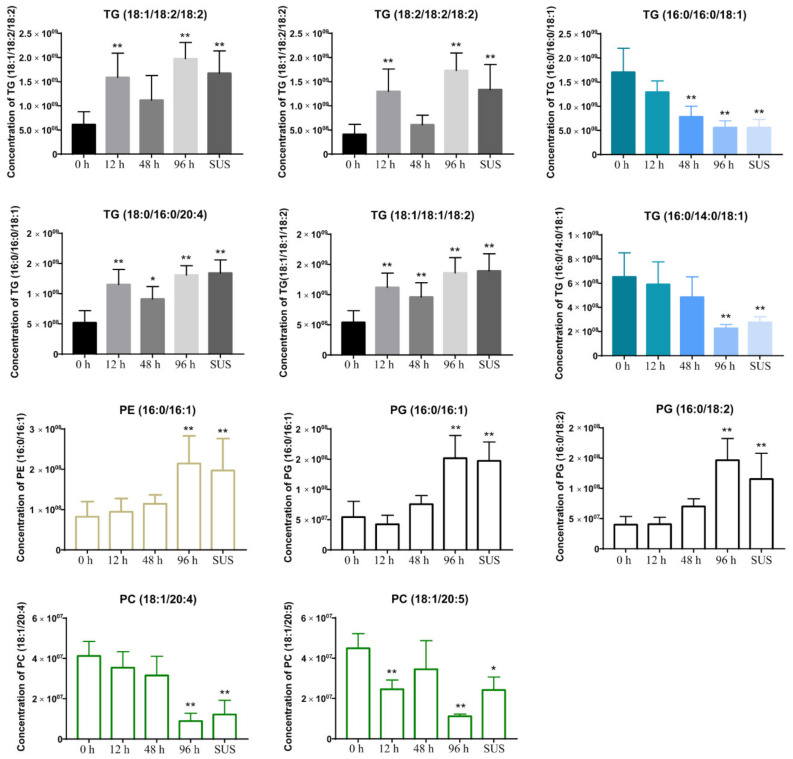
The major significant metabolites throughout the different times after the *V. splendidus* infection. Asterisks indicate significant differences: * *p* < 0.05 and ** *p* < 0.01.

**Figure 7 ijms-23-13583-f007:**
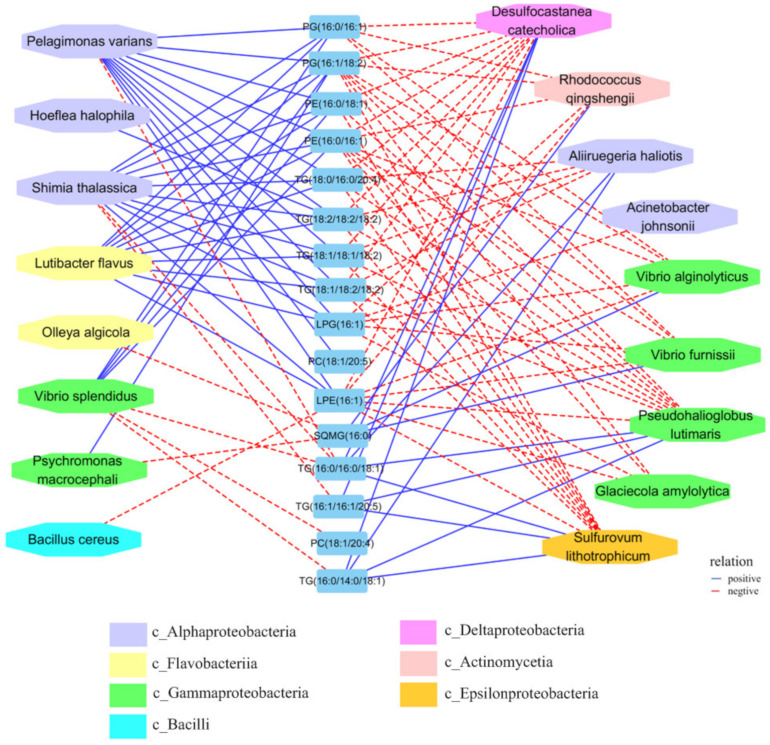
Correlation network analysis of the key lipid metabolites and the major intestinal microbiota throughout the progress of the *V. splendidus* infection. The solid (or dashed) arrows represent positive (or negative) interactions between the two individual taxa. OTUs are colored to indicate the bacterial class level.

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
