# Peer review of "Gut Microbiota Mediates Skin Ulceration Syndrome Outbreak by Readjusting Lipid Metabolism in Apostichopus japonicus"

_ijms, 2022, doi:10.3390/ijms232113583_

Round 1
Reviewer 1 Report
The article "Gut microbiota mediates skin ulceration syndrome outbreak 2 via readjusting lipid metabolism in Apostichopus japonicus" by Zhang et al is interesting. However, there are major flaws in this study.
There is no coherance all over the study. The introduction part is very long. It should be concise and represent the gap between previous studies. It should also give the significance of this study. Moreover, future prespectives are also absent.
The English grammar should be rechecked by a native English speaker to improve the quality of the paper.
The author obtained the sea cucumbers at 0 h, 24 h, 48 h, and 96 h after the post-inoculation of V. splendidus. But in the results, the author mentioned the results of 12 hours instead of 24 hours. Furthermore, there is no clarification why the triglycerides (TG) (18:1/18:2/18:2; 18:2/18:2/18:2; 18:0/16:0/20:4; 18:1/18:1/18:2) increased after 12 hours and then decreased at 48 hours and increased again after 96 h. So it should be explained and so on with other results as well.
In the materials and methods part, the author should split the statistical analysis part in order to understand it more effectively.
The conclusion part should be improved by giving the strategies and recommendations to reduce uncertainties in this study and explaining the implications and future work considering the outputs of the current study.
Author Response
Comments and Suggestions for Authors
The article "Gut microbiota mediates skin ulceration syndrome outbreak via readjusting lipid metabolism in Apostichopus japonicus" by Zhang et al is interesting. However, there are major flaws in this study.
Point 1: There is no coherance all over the study. The introduction part is very long. It should be concise and represent the gap between previous studies. It should also give the significance of this study. Moreover, future perspectives are also absent.
Author response: Thank you for your constructive review. We prune the introduction part in the revised manuscript, and we represent the gap between previous studies and the significance of our study by “Most studies on gut microbiota and metabolic disorders have focused on short-chain fatty acids. Studies on long-chain fatty acids have rarely been reported.”, “Nevertheless, no studies have been published that focus on detecting the changes in the gut microbiota and metabolic patterns before SUS, which may be regarded as biomarkers to predict or prevent disease development.”, and “Together, these findings reveal distinct categories of dysbioses, which can inform the development of microbiome-based diagnostics and therapeutics.”. We add those parts in our re-submitted manuscript at line 44-89.
Point 2: The English grammar should be rechecked by a native English speaker to improve the quality of the paper.
Author response 2 We thank the reviewer for your careful proofreading and recheck the English grammar for entire manuscript by the MDPI english language editing service (ID# 51920).
Point 3: The author obtained the sea cucumbers at 0 h, 24 h, 48 h, and 96 h after the post-inoculation of V. splendidus. But in the results, the author mentioned the results of 12 hours instead of 24 hours. Furthermore, there is no clarification why the triglycerides (TG) (18:1/18:2/18:2; 18:2/18:2/18:2; 18:0/16:0/20:4; 18:1/18:1/18:2) increased after 12 hours and then decreased at 48 hours and increased again after 96 h. So it should be explained and so on with other results as well.
Author response: We are very sorry for our negligence. We obtain the sea cucumbers at 0 h, 12 h, 48 h, 96 h and SUS after the post-inoculation of V. splendidus, and modify the method of 12 h with its consistent with the result. We appreciate the reviewer’s constructive comments. In the process of inflammation induced by pathogen infection, there is a process from acute inflammation to chronic inflammation similar to that of higher animals. Sudden 12 h stimulation leads to the upregulation of inflammation-related substances to clear the bacteria. After 48 h, organism resistance leads to the bacteria not being dominant. At 96 h, the organism cannot maintain bacterial resistance and develops chronic inflammation or traumatic inflammation. For example, TGs (18:1/18:2/18:2; 18:2/18:2/18:2; 18:0/16:0/20:4; 18:1/18:1/18:2) promote inflammation in the branched-chains, which induce inflammation when upregulated, but when downregulated, some branched chain lipids provide basic energy or inhibit inflammation, such as TGs (16:0/16:0/18:1; 16:0/14:0/18:1), showing the same change trend as that of intestinal microbiota in the process of inflammation. We enrich the discussion in our re-submitted manuscript at line 368-378.
Point 4: In the materials and methods part, the author should split the statistical analysis part in order to understand it more effectively.
Author response: We thank the reviewer for this comment. W add “4.7. Statistical analysis All results are presented as the mean ± standard deviation (SD). Whether the data had a normal distribution was verified by Tukey's post hoc test and a one-way analysis of variance test (SPSS, Version 19.0), and the data with p < 0.05 were considered significant.”in the revised materials and methods part.
Point 5: The conclusion part should be improved by giving the strategies and recommendations to reduce uncertainties in this study and explaining the implications and future work considering the outputs of the current study.
Author response: We appreciate the reviewer’s constructive comments. To fully understand the connections between gut microbiota and lipid metabolites, future studies should isolate and culture a single bacterium to address this issue and determine what kinds of long-chain fatty acids are the main factors regulating inflammation. The current work may improve our current understanding of sea cucumber lipid metabolites during pathogen infection. We explain this point in our revised manuscript.
We have revised the manuscript to reflect the reviewers’ suggestions. These changes will not influence the content and framework of the paper. We appreciate the Editor’s/Reviewers’ suggestions and hope that the revised manuscript will meet with approval.
Once again, thank you very much for your comments and suggestions.
Sincerely,
Chenghua Li

Reviewer 2 Report
The manuscript entitle "Gut microbiota mediates skin ulceration syndrome outbreak 2 via readjusting lipid metabolism in Apostichopus japonicus" is very well written and presented. In this is a very interesting theme that can give very interesting data to the field. Overall is this very weel presented and is easy to read and understand.
Author Response
Comments and Suggestions for Authors
The manuscript entitle "Gut microbiota mediates skin ulceration syndrome outbreak 2 via readjusting lipid metabolism in Apostichopus japonicus" is very well written and presented. In this is a very interesting theme that can give very interesting data to the field. Overall is this very weel presented and is easy to read and understand.
Author response: We thank the reviewer for the affirmative comment.

Reviewer 3 Report
The study was targeted to evaluate the influence of gut microflora on skin ulcerations. The authors investigated an expression of pro inflammatory cytokines under there influence of the intestinal microbial community. The study design is clear. The article is well structured and written and could be accepted by the readers with sufficient interest. The only one recommendation for the authors is to extend the Introduction with the most relevant clinical publications on human gut microbiota. The author should also avoid an overloaded sentences (example: Abstract. lines 11-16).
Author Response
Comments and Suggestions for Authors
The study was targeted to evaluate the influence of gut microflora on skin ulcerations. The authors investigated an expression of pro inflammatory cytokines under there influence of the intestinal microbial community. The study design is clear. The article is well structured and written and could be accepted by the readers with sufficient interest. The only one recommendation for the authors is to extend the Introduction with the most relevant clinical publications on human gut microbiota. The author should also avoid an overloaded sentences (example: Abstract. lines 11-16).
Author response: We appreciate the reviewer’s constructive comments. We describe the gut microbiota as bacterial markers and the possible clinical publications in the introduction. The previous studies showed that Blautia、Faecalibacterium and Ruminococcus were the major maleficent bacteria of crohn disease and ulcerative colitis (Yilmaz et al., 2019; Schirmer et al., 2019), and Clostridium immunis was discovered that-when administered to colitis-prone mice-protected them against colitis-associated death (Surana and Kasper, 2017). In addition, studies have pinpointed the roles of several bacteria in colorectal carcinogenesis, including Fusobacterium nucleatum and certain strains of Escherichia coli and Bacteroides fragilis, and reported associations between bacterial markers and treatment efficacies or clinical outcomes, increasing the potential of using these markers for treatment prediction and prognostication (Wong et al., 2019; Behary et al., 2021). We add this point in line 61-69, and make our abstract part to be concise, especially line 11-16.
We have revised the manuscript to reflect the reviewers’ suggestions. These changes will not influence the content and framework of the paper. We appreciate the Editor’s/Reviewers’ suggestions and hope that the revised manuscript will meet with approval.
Once again, thank you very much for your comments and suggestions.
Sincerely,
Chenghua Li

Round 2
Reviewer 1 Report
Now the present paper is ready for publication